# Chitosan- and Alginate-Based Hydrogels for the Adsorption of Anionic and Cationic Dyes from Water

**DOI:** 10.3390/polym14081498

**Published:** 2022-04-07

**Authors:** Mohammad T. ALSamman, Julio Sánchez

**Affiliations:** Departamento de Ciencias del Ambiente, Facultad de Química y Biología, Universidad de Santiago de Chile (USACH), Santiago 9170022, Chile; mohammad.alsamman@usach.cl

**Keywords:** adsorption, alginate, chitosan, methylene blue, methyl orange, water treatment

## Abstract

Novel hydrogel systems based on polyacrylamide/chitosan (PAAM/chitosan) or polyacrylic acid/alginate (PAA/alginate) were prepared, characterized, and applied to reduce the concentrations of dyes in water. These hydrogels were synthetized via a semi-interpenetrating polymer network (semi-IPN) and then characterized by Fourier transformed infrared spectroscopy (FTIR) and thermogravimetric analysis (TGA), and their swelling capacities in water were measured. In the adsorption experiments, methylene blue (MB) was used as a cationic dye, and methyl orange (MO) was used as an anionic dye. The study was carried out using a successive batch method for the dye absorption process and an equilibrium system to investigate the adsorption of MO on PAAM/chitosan hydrogels and MB on PAA/alginate in separate experiments. The results showed that the target hydrogels were synthetized with high yield (more than 90%). The chemical structure of the hydrogels was corroborated by FTIR, and their high thermal stability was verified by TGA. The absorption of the MO dye was higher at pH 3.0 using PAAM/chitosan, and it had the ability to remove 43% of MO within 10 min using 0.05 g of hydrogel. The presence of interfering salts resulted in a 20–60% decrease in the absorption of MO. On the other hand, the absorption of the MB dye was higher at pH 8.5 using PAA/alginate, and it had the ability to remove 96% of MB within 10 min using 0.05 g of hydrogel, and its removal capacity was stable for interfering salts.

## 1. Introduction

Chitosan is a positively charged polysaccharide and is frequently used as a sorbent to remove methyl orange (MO) from aqueous solutions by adsorption. Combined experiments have been conducted to study the effects of adsorption, and chitosan has proven highly effective in the adsorption of anionic dyes, with a negative charge due to its positive charge [1]. Chitosan and its derivatives have been widely used as sorbents for various types of pollutants in water, where they can be used to obtain composite materials, nanocomposites, hydrogels, and membranes, and combined with natural elements such as clay, saccharides, and as well synthetic polymers. These materials can be non-covalently assembled depending on the hydrogen bonds, van der Waals, and π–π donor–acceptor, and there are types like covalently crosslinked hydrogel materials and others such as hybrid hydrogels [2,3]. In addition, chitosan can be added to inorganic materials such as zeolite to form aerogels with adsorption performance for anionic and cationic dyes. The molecular interaction between chitosan and the zeolite can produce a steric effect, and their morphology implications might play a role in the aerogel and their active site, leading to better adsorption [4].

Chitosan is a natural biopolymer processed from chitin. It has received considerable attention because it is nontoxic, renewable, and biocompatible. However, it has some drawbacks such as insufficient stability in an acidic environment, and its poor mechanical and thermal strength, particle size, and surface area restrict its industrial use. To overcome these drawbacks and enhance its adsorption potential, chitosan is often physically and chemically modified. For example, chitosan has been modified via a semi-interpenetrating polymer network (semi-IPN) with synthetic polymers such as polyacrylamide (PAAM) [5,6]. Zhao and his collaborators created semi-IPN hydrogels, which were prepared for dye uptake studies by photopolymerization of poly(ethylene glycol) and acrylamide monomer in the presence of chitosan down to a Qmax of 202.02 mg/g based on the Langmuir adsorption model for MO dye [7]. In addition, Cheng Zhang and collaborators created polyacrylamide/chitosan (PAAm/CS) and Fe_3_O_4_ hydrogels, with *N*,*N*′-methylene bis-acrylamide as a crosslinker, to adsorb methylene blue (MB). The results showed that the PAAm/CS hydrogels and PAAm/CS/Fe_3_O_4_ hydrogels can both adsorb MB, but the presence of Fe_3_O_4_ increased the adsorption capacities to approximately 1603 mg/g [8]. Alginates are naturally occurring polysaccharides that have many applications, including the removal of pollutants from water. They contain linear structures and consist of two types of residues linked in a 1→4 manner. One of the monosaccharide residues is β-d-mannuronic acid (M), and the other is a C-5 epimer, α-l-guluronic acid (G). Alginates contain carboxyl groups (protonated or ionized), which provide ionic strength that helps remove contaminants [9]. Hence, sodium alginate contains a large number of -COO- groups, and alginate hydrogels also act as natural biodegradable absorbent materials for industrial wastewater treatment, especially for dye pollutants, and they can avoid secondary pollution or other environmental issues [10]. Upon physical or chemical modification of gels, the technological properties of the final product are largely subject to alginate variables such as chemical structure, sequence, molecular weight, and molecular weight distribution. Important technological properties are gel strength, porosity/diffusion, alginate distribution, swelling/shrinkage, transparency, and alginate leaching from gels, all of which must be optimized for different systems depending on their application [11]. Polysaccharides are among the materials that reduce mechanical aging as it occurs at the lowest temperatures such as in the aging behavior of elastomer under high temperatures and pressure, and thermal oxidation and thermo-hydrolytic conditions can also cause a kind of resistance when adding polysaccharides to the process of deformation in torsion by “zero-time” response and increase of the step strain [12,13]. Maijan and collaborators created hydrogels that were developed on the basis of natural rubber (NR). NR was concomitantly combined with the NR-graft-polyacrylamide (NR-g-PAM) hydrogel to form a freely movable secondary polymer network via physical chain crosslinking. A semi-IPN was formed, and 30NRgraft-polyacrylamide removed 90% of MB from an aqueous solution in less than 24 h, with a maximum absorption capacity of 538.3 mg g/g [14]. Hosseinzadeh and collaborators created a new material prepared using acrylic acid (AA) grafting on sodium alginate (NaAlg), with ammonium persulfate as a free radical initiator and methylene bisacrylamide as a crosslinker, in the presence of SiO2-NP, to absorb MB, revealing a capacity of 148.23 mg/g [15]. In this study, hydrogels were prepared with PAAM/chitosan and PAA/alginate, and their ability to remove dyes from water was determined through batch experiments. The swelling in water, FTIR spectra, and TGA characterization, and the removal of dyes as a function of the biopolymer effect, dose, pH, contact time, concentration of dyes, and salt concentration, were studied.

## 2. Materials and Methods

Chitosan (85% deacetylation) CAS: 9012.76.4 (Sigma-Aldrich, Darmstadt, Germany), sodium alginate (90% carboxylation) CAS: 9005.38.3 Merck (Sigma-Aldrich, Shanghai, China), acrylamide CAS: 76.06.1 (Sigma-Aldrich, Darmstadt, Germany), acrylic acid CAS: 79.10.7 (Sigma-Aldrich, Darmstadt, Germany), *N*,*N*-methylenebisacrylamide (MBA) CAS: 110.26.9 (Sigma-Aldrich, St. Louis, MO, USA), ammonium persulfate (APS) CAS: 7727.54.0 (Sigma-Aldrich, Darmstadt, Germany), sodium hydroxide CAS: 1310-73-2 (Sigma-Aldrich, Darmstadt, Germany), lithium carbonate CAS: 554-13-2 (Sigma-Aldrich, Darmstadt, Germany), lithium chloride CAS 7447-41-8 (Sigma-Aldrich, Darmstadt, Germany), calcium carbonate Cas No: 471-34-1 (Sigma-Aldrich, Darmstadt, Germany), sodium chloride CAS 7647-14-5 (Sigma-Aldrich, Darmstadt, Germany), potassium chloride CAS 7447-40-7 (Sigma-Aldrich, Darmstadt, Germany), hydrochloric acid solution CAS No: 7647-01-0 (Sigma-Aldrich, Darmstadt, Germany), methylene blue hydrate CAS: 122965-43-9 (Sigma-Aldrich, Darmstadt, Germany), and methyl orange CAS: 547-58-0 (Sigma-Aldrich, Darmstadt, Germany), were all purchased from Merck Group Francisco de Paula 1981, Ñuñoa, (Merck S.A. Sigma–Aldrich, Región Metropolitana, Chile).

### 2.1. Synthesis of Hydrogel via Semi-IPNs Using Free Radical Polymerization Reactions

The semi-IPNs of both PAA/alginate and PAAM/chitosan were both synthesized from their monomers (see Figure 1) by free radical polymerization in a Schlenk tube reactor in a silicon bath at 70 °C for 1 h using APS as the redox pair initiator and MBA as the crosslinker [16]. 

First, we added monomer powders while varying the amount of crosslinking agent (0.25, 2.5, 10, 24, and 60%) and varying the biopolymer amounts to create different ratios and concentrations of the prepared hydrogels, as shown in Figure 2 and Table 1. 

After the preparation of the hydrogels, they were washed and stored under refrigeration and then lyophilized to remove unreacted monomers and preserve the constituent charges of the hydrogel, after which dry xerogels were obtained.

### 2.2. Characterization of the Hydrogel by Swelling Capacity, FTIR, and TGA

The dynamic swelling properties of the hydrogels were determined at different times, and they were studied to determine their ability to absorb water and remain stable. The swelling ratio (*St*) and the equilibrium swelling ratio (*Se*) were gravimetrically calculated using Equations (1) and (2) [17].
(1)$$Swelling\;ratio\;\left(St\right)=\frac{Wt-Wi}{Wi}$$
(2)$$Swelling\;\left(Se\right)=\frac{We-Wi}{Wi}$$
where *Wi* is the initial weight of a dry hydrogel sample, *Wt* is the weight of the inflated sample at time *t*, and We is the weight at equilibrium when there is no further change in weight with time (see Equations (1) and (2)) [18]. FTIR was performed using a Perkin-Elmer FTIR Spectrum coupled to an ATR unit. The dry sample was directly placed over the diamond, pressed up to 30% underpin pressure, and then analyzed via FT-IR scanning in the range of 500 to 4000 cm^−1^ [19]. TGAs were conducted using the TG 209 F1 system by IRIS, NETZSCH brand. The samples were placed inside a capsule made of aluminum followed by a process of exposing them to heat, and the temperature was increased up to 500 °C in an inert atmosphere of nitrogen [20].

### 2.3. Removal of Dyes

The removal efficiency was calculated as follows:(3)$$Removal\;Efficiency=\frac{\left(C_{0}-C_{t}\right)\;}{C_{0}}\times 100\%$$

Additionally, the adsorption capacity of dyes was calculated as shown below
(4)$$qt=\frac{\left(C_{0}-C_{t}\right)\;V}{m}$$
where *C*_0_ is the solution concentration at the initial time, *C_t_* is the concentration at equilibrium (mg/L), *m* (g) is the mass of adsorbent, and *V* (mL) is the volume of the solution [21]. These equations were used to study the effect of parameters such as the biopolymer effect, dose, pH, contact time, concentration of dyes, and interfering salt concentrations on dye removal by the xerogels. Triplicate experiments were performed for all operating variables studied, and results were reported as the mean values. It was found that the average deviation of the results was low [22]. The batch dye removal experiment was carried out in a centrifuge for 6–12 min at 9000 rpm for MB and 4000 rpm for MO, and absorption was measured using a UV spectrophotometer UV–vis with a wavelength of 464 nm for MO and a wavelength of 664 nm for MB. The dye removal efficiency was calculated as follows and according to Equations (3) and (4) [23,24], where the variables were the value of the weight of xerogels and biopolymer quantities to verify the best proportion of biopolymer. Where the effect of the biopolymer amount was studied, an amount of 0.05 g was placed in sample tubes that contained different biopolymer percentages (0, 3, 7, 12, 24, 60%), and the results showed that 24% was the best amount to use for adsorption of the model dyes. A study was conducted to determine the optimal pH by placing 0.05 g of gels and 10 mg/L of dyes in 10 mL tubes into the shaker device for 90 min at 200 rpm, and then the measurement was performed with UV equipment. Where the effect of pH was studied, methylene blue dye was placed in different pH media (pH = 2, 6, 8, 8.5, 9, and 10), and a decimal study was conducted from 8 to 9. Similarly, the effect of pH was studied for MO dye adsorption at pH values of 2, 3, 4, 6, and 10, with a decimal study from 2 to 3.5. To study the effect of the concentrations of dyes on their removal, the process was conducted as detailed in the previous steps, but the pH was fixed and the concentrations were varied (10, 50, 100, 200, 300, 400, 500, 600, 800, 1000, 1300, and 1500 mg/L). In the case of MO, the maximum concentration was 1000 mg/L. Dye adsorption was also studied via batch experiments at different contact times; 10, 20, 30, 60, 90, 120, and 240 min experiments were carried out in tubes at room temperature using an MB solution at a pH of 8.5 to study the effect of contact time on MB adsorption on the biobased hydrogels. To study the effect of interfering salts on dye removal, salts were added at 1:1, 1:2, and 1:3 dye:interfering ion molar ratios, and the previous experimental procedures were applied at a concentration of 100 mg/L for MO and 500 mg/L for MB.

## 3. Results and Discussion

### 3.1. Swelling Capacity of Hydrogels in Water

The swelling of hydrogels is modulated by osmotic pressure, where large amounts of water initially enter the hydrogel structure and form hydrogen bonds, and the gradual diffusion of more water into the hydrogel network then leads to a difference in osmotic pressure between the hydrogel system and the surrounding water, reducing the opposing elastic contraction force [25]. Since the movement of mobile charges from the hydrogel network is restricted to the surrounding water, when it reaches a state of equilibrium, the osmotic pressure of the opposite force and the elastic contraction are equal, and no new swelling occurs because the hydrogel network reaches an elongated formation [26]. The polymer network can retain water as the network crosslinking the polymeric chains reaches the equilibrium stage by weakening the hydrogen bond interaction between the hydrophilic groups. These groups increase the electrolyte concentration in the inner gel network and enhance the osmotic pressure difference between the gel network and the swelling media but also enhance the repulsion between the charged polymer chains. The enhanced monomer concentration increases the diffusion of monomer molecules into the hydrogel network and thus causes increased water absorption [27]. The swelling of the hydrogels prepared with 24% biopolymer was higher than that of the blank (0% biopolymer), and the results indicated that the water absorption capacity increased initially with increasing ion concentration and then decreased (see Figure 3). Theoretically, with the increase of molecular weight of chitosan or alginate after 20 min, its swelling value reaches a higher limit due to the larger spacing between the cross-linkers and the resulting hydrogel having a larger network size, so it reaches a higher swelling rate and retains water within the network of gels without dissolving. The swelling gradually and progressively increased in proportion to the percentage of the biopolymer. In the case of chitosan hydrogels, there was an upwards increase, but the values were similar in gels containing low levels of chitosan. The swelling was higher than 500% in the case of chitosan and 450% in the case of alginate. The xerogels (dried gels) containing chitosan and polyacrylamide are higher in swelling than the polymer, and the swelling percentage increases with increasing chitosan concentration, which makes PAAM/chitosan have a 24% higher swelling rate than PAA/alginate. These gels have better performance and higher swelling because the one that contains chitosan contains more amine groups, which results in more hydrophilic groups, such as OCOO and OCONH2, whereas PAA/alginate only contains OH and COOH groups and only forms hydrogen bonds, resulting in less swelling than its counterpart [28,29,30]. The initial increase in the swelling capacity values was due to the high degree of crosslinking in the polymeric chains [31]. It showed a higher swelling ratio due to the relatively low degree of cross-linking compared to other hybrid hydrogels, where hydrogen bonding between the carboxylic acid and amide groups at lower pH values reduced the areas of the hydrogel network and consequently resulted in lower swelling rates [32]. The PAA/alginate gel has many hydroxyl (-OH) groups, and the COOH groups in its structure might lead to hydrogen bonding interactions among these groups, which is caused by the repulsion of those groups from each other. Then, these groups would expand the gel network [33]. On the other hand, PAAM/chitosan hydrogels were significantly superior to PAA/alginate hydrogels. This increase was associated with the presence of many hydrophilic groups, such as amine groups, in PAAM composite gels [34].

### 3.2. Characterization by FTIR

The spectra of PAAM/CH (0% to 24%) (see Figure 4) showed vibrations that represent the crosslinked polyacrylic amide, and the peaks appear at 2947 cm^−1^ (amide 1) and 1732 cm^−1^ due to C=O stretching vibrations and the range between 1000 and 1500 cm^−1^. These bands represent vibrations of C–H, C–CH, and N-H bending (1139, 1232, and 1457 cm^−1^) [35]. The addition of chitosan imparts crosslinking and forms crosslinked chitosan. The amino bonds disappear, the amide bonds are formed, and they appear in the spectra. At the wavelength of 2952 cm^−1^, the amino bonds disappear, and when bonds are formed, an amide is formed and vibration bands appear at 956 to 1756 cm^−1^ due to the saccharide structure (due to O-H bending, C-O stretching, and C-N stretching).

The band appearing in the wavenumber field 2800–3500 cm^−1^ could be assigned to the stretching vibration of O–H and the extension vibration of N–H and was caused by the interhydrogen bonds of the absorption peaks at 1500 and 1700 cm^−1^ corresponding to its carboxylic C=O stretching at 1657 cm^−1^, which was due to its protonated amine (NH^4+^) groups [36,37]. Additionally, the FTIR spectrum showed additional bands of (N-H) stretching at 1225 cm^−1^ (amide 1) and 1375 cm^−1^ (amide II). The combined band of O-H and N-H of (amide I and amide II) stretching is seen at 2800 to 3800 cm^−1^, and the band at 3310 cm^−1^ is due to its hydroxyl (O-H) group [38].

The FTIR spectra of PAA/alginate presented in Figure 5 were analyzed by comparing the difference between spectra obtained from samples with different amounts of alginate [39]. When no sodium alginate is present and the hydrogel sample is only composed of cross-linked PAA, the peaks representing cross-linked PAA are in the absorption range of 1350 to 1850 cm^−1^. The range is 1700 cm^−1^, and the shoulders at 1710 and 1765 cm^−1^ are due to the vibration of the carboxylate expansion C=O. The bands with peak locations at 1541, 1454, and 1378 cm^−1^ are the vibrations of COH bending of PAA. The range at 1700 cm^−1^ with shoulders at 1639 and 1705 cm^−1^ is attributed to the C=O expansion vibration of carboxylic PAA, which is proportional to the PAA concentration, and hydrogen bonds are the cause of the bulging in the peaks [40]. When the biopolymer concentrations differ, we note that there are no absorption bands at 1706 cm^−1^ (asymmetric COO) and 1235 cm^−1^ (symmetric COO) changing at frequencies lower than 1750 and 1277 cm^−1^, and we conclude that free radical polymerization does not affect the individual characteristics of the polymer [41]. Furthermore, we observed that the absorption of the C-O-C group of the anhydride appeared to be successful. There are other domains affected by anhydride formation, and they can be assigned to C-O or O-H vibrations, sometimes coupled to C-H oscillations, which are the causes of the hydrogen bonding causing the oscillations [42]. The very strong characteristic band at 1706 cm^−1^ is due to stretching in the carboxylate anion, which is confirmed by another sharp peak at 1172 cm^−1^, which is related to the symmetric stretching mode of the carboxylate anion. The FTIR spectrum of the hydrogel hybrid was also compared to the spectra of the partial hydrogel [43]. We observe that the peaks presented in the figure are similar, while the appearance of a large amount of sodium alginate and its high concentration leads to the appearance of three peaks that express the formula of the functional group showing an absorption range of 1141 and 1268 cm^−1^ [44], due to the extended vibration of the C-OH group and 1644 cm^−1,^ corresponding to the vibration of asymmetric expansion of its carboxylic group (COO). As the hydrogel creates a layer, this layer covers and reduces the hydrogen bonds; that is, it becomes very soft due to the OH and -COO- bonds [45]. We noticed the disappearance of the peak from 2310 to 3700 as the proportion of the biopolymer increased, which is evidence of the formation of semi-IPN and the emergence of etheric bonds and the disappearance of OH groups.

### 3.3. Characterization by TGA

Thermogravimetric analysis (TGA) was used to characterize the thermal stability of the PAAM/chitosan and PAA/alginate hydrogels. The samples were studied in the temperature range from 0 °C to 500 °C under nitrogen [46,47].

Figure 6A shows the TGA curves for PAAM. In the first region, occurring between 64 and 299 °C, weight is lost with temperature up to 100 °C due to evaporation of free water and water bound via hydrogen bonds. The backbone of the polymer can be nonionic (-OH, -O-, -NH_2_, -CONH-, and -CHO) or ionic (-COOH, -COONa, COONH_4_, etc.). At 121 °C, the slight weight loss of 11.2% was attributed to the decomposition of chitosan. The degradation of the chitosan is caused by the degradation of the main chain followed by non-isothermal kinetics, which contains hydrophilic function groups, from which the elements of water are removed. The second zone starts at 300 °C and ends at 349 °C, with a maximum decomposition rate at 301 °C and a weight loss of 7.8%. Then, the third zone, above 350 °C, has a maximum decomposition rate at 434 °C and a weight loss of 42.1%, with a maximum decomposition rate at 456 °C and a weight loss of 46.43% by weight between 270 and 320 °C, and it lost approximately between 42.1 and 48.1% of its weight. The observed behaviors can be attributed to the oxidative degradation of chitosan backbones at different concentrations that are assigned to the heat decomposition of organic molecules. This decomposition is associated with a complex process of drying the polysaccharide rings, and depolymerization and decomposition of the acetylated and deacetylated chitosan units; the fourth zone ranges from 436 °C to as much as 500 °C, resulting in decomposition and ash [48,49,50,51,52,53,54,55,56]. 

Figure 6B shows a weight loss of 2.5% in the first region (between 0–243 °C), with a maximum decomposition rate of 111 °C, involving decomposition of the COOH and OH groups as non-radical back biting ester interchange reaction for alginate involving the -OH chain ends and release of water molecules for uptake and diffusion. We noticed that when the temperature increases, these groups disintegrate and release water; therefore, we noticed rapid displacement, disintegration, and thermal instability compared to chitosan gels. The second region starts at 234 °C, reflecting thermal oxidative decomposition of organic matter and acid liberation of acrylic groups, indicating decompression; it persists up to 345.5 °C, with a weight loss of 15.5%, and the third region of 346 °C–358 °C can be attributed to the degradation of the corresponding PAA anhydride. The fourth region is at 436 °C and could be due to thermal degradation of PAA, and the main weight loss occurs in the temperature range of 400 °C to 450 °C. Overall, the PAA/alginate hydrogel exhibited high thermal stability at various heating temperatures, with minimal weight loss [57,58,59,60]. The thermal stability of chitosan hydrogels is due to bonds formed between O-H, amide, and C-N groups. It is known that this is a strong bond that provides stability towards thermal fluctuations, and therefore chitosan gels have more stability due to the strength of this bond. On the other hand, alginate hydrogels have thermal instability due to carboxylic functional groups being easily converted to carbon dioxide and water, which causes rapid thermal degradation; therefore, alginate gels are thermally unstable.

### 3.4. Removal of Dyes

#### 3.4.1. Effect of Biopolymer Percentage on Dye Removal

It was confirmed that increasing the amount of chitosan causes an increase in the number of active sites, adds to the adsorption effectiveness, and increases the adsorption value of MO at pH 3 (see Figure 7A), but to a certain extent, acrylamide imparts stability properties to the sample and causes an increase in stability against the dye. The results show an increase in dye adsorption using hydrogels with 24% biopolymer. Additionally, we note that percentages such as 60% have the same absorption value as hydrogels with 24% biopolymer [61]. 

The adsorption of MB on xerogels was studied by varying the alginate percentage at an MB concentration of 10 mg/L and a pH of 8.5 (see Figure 7B). According to Figure 7B, we note that the value of the dye removal increases with the increase in the organic polymer because it is an organic polymer with reactive hydrogens on its chains, usually closely related to electrostatic attraction and hydrogen bonds, according to previous studies. In addition, the alginates form COO- groups, which greatly enhance the electrostatic attraction between the alginate compounds and the dye molecules [62,63].

#### 3.4.2. Effect of the Amount of Adsorbent on Dye Removal

Figure 8 shows that during the use of 0.0125 g of hydrogel; the value q (mg/g) was low and did not enable one to remove dyes at high concentrations, as was the case with 0.0250 g of hydrogel. However, when using 0.05 g of hydrogel, the q value increased, and it became relatively high. Additionally, it maintained a stable q (mg/g) value, and there was no increase between 0.01 g and 0.015 g of gel. In the case of chitosan gels removing MO (see Figure 8A), we noted that when the weight increases, it reaches a stable state at 0.05 g of gel. This is the most suitable weight for a volume of 10 mL and has an effective absorption capacity of 66%. There is a proportional relationship between the amount of chitosan and absorption, and increasing amounts of chitosan results in an increase in electrostatic bonding, which increases the absorption of dyes. This can be attributed to the increased adsorption surface area and availability of more adsorption sites because the carboxylic group of acrylic acid and sodium alginate have the capacity to adsorb cationic dyes [64,65]. As these saccharides have high abundance and come from natural sources with good adsorption characteristics, they can be used with industrial polymers such as acrylic acid. Such combinations of alginates and industrial polymers can be used to adsorb cationic dyes that carry a positive charge at an optimum amount of 0.05 g, as shown in Figure 8B. We noticed that there is no increase in absorption above this amount; it is the most appropriate weight for a volume of 20 mL, and it has a capacity of 96% effective absorption. We noted that when the percentage of alginates increases, the absorption increases, and there is an increase in electrostatic bonding, which increases the adsorption of these dyes [66,67,68].

#### 3.4.3. Effect of pH on Dye Removal

The effect of solution pH on adsorption of MO dye on chitosan gels was initially studied in the pH range of 2.0–10.0. Figure 9 shows the effect of pH on adsorption efficiency. Chitosan has a positive charge, providing it with high potential for adsorption of MO dye. Where the pH was low, the adsorption efficiency was high as the charged state of the MO dye was affected by the pH of the solution. At low pH values, the solution causes electrostatic attraction between MO and chitosan, and the absorption increases [69]. Positive charge predominates on the surface (i.e., -OH_2_^+^, and -NH_3_^+^). In contrast, the presence of “NH_2_” (i.e., electron-donating functional groups) on the surface of chitosan is a contributor to the adsorption of MO (see Figure 9A). The adsorption efficiency of MO (pKa = 3.46) at pH = 3.0 is attributed to the strong electrostatic forces between the positively charged sites (functional groups) of the PAAM/chitosan hydrogel and the negatively charged anionic MO. Another factor in the absorption of MO at pH 3.0 is hydrogen bonding. The adsorption of MO was lower at basic pH due to the lower positive charge, which causes MO functional groups to be associated with different interactions, such as n–π interactions, in MO adsorption [70,71,72]. The main mechanism explaining the absorption of MO is electrostatic interactions. When the absorbent material has a positive charge, as chitosan does, an acidic solution that contains a large positive charge leads to the attraction of the negative molecules of MO dye. The hydrogen atoms can form bonds between the sorbent molecule and the dye, where the highest degree of absorption was obtained at low pH, which leads to electrostatic attractions due to the positive charge due to chitosan amine groups being transformed by protonation at low pH. Because acidic media contributed to the positive charge on chitosan, a pH of 3.0 was chosen for batch studies [73,74,75]. The results show that the equilibrium adsorption capacity as a function of pH expresses an important basic factor in the adsorption of the dye [76,77,78,79,80,81]. On the other hand, there is an electrostatic repulsion between similar ionic charges, i.e., diffusion of H+ in the acidic medium and MB cations, so low pH hinders the adsorption of MB due to the filling of the MB cations. The cationic dye (MB) competes for adsorption sites due to the large number of H + ions in the solution where the number of positively charged sites decreases with the increase in pH, the negatively charged OH- ion levels rise (due to deprotonation), the adsorption of these positively charged dyes increases, and the removal efficiency (MB) improves. This is due to the presence of carboxyl (COO) and hydroxyl (OH) groups in the negatively charged alginate gels in the solution where the surface of the adsorbent becomes more negatively charged, thus enabling the removal of MB [82,83,84] (see Figure 9B).

#### 3.4.4. Effect of Contact Time

The removal of MO dye on chitosan-based hydrogels was studied (see Figure 10A). The role of MO dye uptake as a function of contact time was studied in samples of chitosan gels from 10–240 min. The initial absorption of MO dye was high due to the presence of more functional sites available on the surface, and it reached equilibrium in 10 min. It remained constant after that, and it can be explained that MO dyes are saturated periodically due to electrostatic bonding, and the adsorption capacity increases up to 10 min Q10 min (13.3, 49.5 mg/g) for MO. After equilibrium is reached, and with any increase in time, the absorption value is a constant value and is in equilibrium because the state of dynamic equilibrium has reached Q24 h (14, 54.3 mg/g) for MO [85,86]. Figure 10B shows that alginates have a great ability to absorb MB dye because the alginate gel is a very fast absorbent for this dye. It absorbs them effectively within 10 min, which is considered a fast time because, after 10 min, the process is stable Q10 min (1.8, 19.3, and 105 mg/g) for MB and reaches equilibrium, and there is no increase in absorption Q24 h (1.9, 19.5, and 107 mg/g) for MB. This is attributed to the abundance of unoccupied sites on the surface of the adsorbent that are accessible for rapid adhesion of MB dye molecules, which may explain the rapid adsorption during the initial contact time [87], where a phase transition occurs, during which the dye molecules occupy empty active sites on the surface of the adsorbent, slowing down adsorption and leading to a repulsive force between the adsorbent on the adsorbent surface and the bulk phase in solution [88]. After a reaction time of 10 min, little change in dye removal was observed, and the removal percentage remained at 90%. Then, the dye was removed from the absorbent and remained stable, with little change with increasing time [89].

#### 3.4.5. Effect of Initial Dye Concentration

MO removal increased when the amount of chitosan was increased from 0% to 24% (see Figure 11A). The increase in the removal percentage was probably due to the increase in the available adsorption surface and the availability of more adsorption sites due to the functional groups of chitosan. It reached a certain limit, and any further increase in MO concentration did not increase the absorption [90].

This may be because the initial MO concentrations provide great strength to overcome all the mass transfer resistance of the dye between the aqueous phase and the solid phase and then reach an equilibrium. It can be confirmed that all the active adsorption sites become saturated after the MO concentrations (10, 50, 100, 200, 300, 400, 500, 800, and 1000 mg/L), reaching Q values (2.5, 9.8, 18.1, 38.5, 49.7, 64, 73, 79, and 81 mg/g) for 0.05 g of PAAM/chitosan 24% at pH 3.0. The absorption of MO in 24% chitosan gels is higher than that of 0% chitosan gels because the surface area and pore volume of the former are higher than those of the latter, and due to the presence of functional groups [91]. The adsorption of MB on xerogels at different concentrations was studied by varying the MB concentration. When the dye concentrations increased, the absorption increased in the dried xerogel that contained alginate (see Figure 11B) [92].

#### 3.4.6. Effect of Interfering Salts

To study the effect of interfering species on dye removal, the following salts were added: LiCl, CaCO_3_, LiCO_3_, NaCl, and KCl. For MB adsorption, the solution pH was 8.5, and Ca^2+^, Li^+^, Na ^+,^ and K^+^ ions were present in the aqueous solution. The MB absorption percentage decreased due to competition with these positively charged ions. Sodium ions are much smaller than K^+^ ions, and the decrease in dye adsorption is more significant in the presence of Na^+^ than K^+^. Ca^2+^ ions compete for positions on functional groups that contain a negative charge, so they block sites due to their active size [93]. Lithium exhibits different behavior because of its small size. The MO dye is different because it is in an acidic medium with a pH of 3.0, where the ions dissociate into Cl^−^ and CO_3_^2−^ and compete for the positive sites on the functional groups of chitosan gels, and the dye absorption decreases due to this competition. It is possible to form salts with MO and cause its precipitation, and we noticed an increase in absorption due to the precipitation of dyes [94]. The effects of ionic strength possess some importance when studying dye adsorption on NaCl and CaCl_2_ adsorbents. It was observed that the increase in salt concentration led to a decrease in the adsorption of MO and MB dyes on the gels and a decrease in the efficiency. This trend indicates the competitive effect between salt ions and cations at the sites available for the adsorption process. With increasing salt concentration, the amount of MO absorbed on the gels (qe) decreased from 18.3 to 14.7, 9.61, 15, 7.68, and 11.8 mg/g for LiCl, Li_2_CO_3_, CaCO_3_, NaCl, and KCl, respectively (see Figure 12A), and the amount of MB absorbed on the gels (qe) decreased from 104 to 99.66, 90.3, 90.54, 95.86, and 98.2 mg/g for LiCl, Li_2_CO_3_, CaCO_3_, NaCl, and KCl, respectively (see Figure 12B).

With the increase in salt concentration from 1:1 mol to 1:2 mol, the amount of MO absorbed on the gels (qe) decreased to 2.23, 4.63, 9.61, 11.8, and 11.23 mg/g for LiCl, Li_2_CO_3_, CaCO_3_, NaCl, and KCl, respectively, and the amount of MB absorbed on the gels (qe) decreased from 94.17, 92.7, 93.62, 94.13, and 100.16 mg/g for LiCl, Li_2_CO_3_, CaCO_3_, NaCl, and KCl, respectively. While it decreased with increasing salt concentration from 1:2 mol to 1:3 mol, the amount of MO absorbed on the gels (qe) decreased to 1.82, 1.65, 3.35, 11.77, and 11.11 mg/g for LiCl, Li_2_CO_3_, CaCO_3_, NaCl, and KCl, respectively, and the amount of MB absorbed on the gels (qe) decreased from 97.4, 92.67, 88.85, 100.60, and 100.83 mg/g for LiCl, Li_2_CO_3_, CaCO_3_, NaCl, and KCl, respectively. One reason for this is that as the ionic strength increases, MB and MO adsorption sites decrease, and thus the MB adsorption capacity decreases. Since Ca^2+^ makes a greater contribution to ionic strength and has more positive charge than Na^+^, the effect of Ca^2+^ on adsorption is stronger than that of Na^+^ [95,96,97,98]. The effect of salt on reducing the adsorption capacity appeared to be due to both the steric hindrance effect and the obstruction of aggregation of particles exerted by the salts, which prevented the formation of adsorption layers and reduced their overall adsorption capacity [98]. The ionic strength present in the solution affected the adsorption of MO or MB on the gels. It was observed that the increase in the salt concentration led to an increase in the absorption and removal efficiency ratio due to the effects of ionic strength or pigment deposition in some cases, as in the case of NaCl with MO pigment with PAAM/CH 0% (see Figure 12C,D) [99,100,101].

### 3.5. Advantage and Disadvantage

Hydrogels have been used for the absorption process; every process has obstacles that stand in its way; and there are benefits resulting from its use, including the fact that high capacity can be applied to many pollutants. Moreover, the fast kinetic development degradation of huge amount of contamination of water bodies are effective and reliable for dye removal, see Table 2 [102,103].

In particular, adsorption technologies are often identified as some of the most prominent strategies, with practical advantages and economical properties for treating water contaminated with dye. They are considered as important hydrophilic materials for adsorption. Chitosan and alginate gels are suitable, unconventional, low-cost materials. These dye-absorbent absorbents are effective for removing a wide range of type of dyes of different charges or other contaminants, are of high capacity and rate of absorption, contain high selectivity for different concentrations, and are capable of a wide range of wastewater parameters [104,105]. Of course, they have disadvantages such as weak mechanical strength and poor absorption at different pH levels [106].

However, chitosan and alginates enhance adsorption efficiency, an effective strategy to prepare polysaccharide-based hydrogels for water treatment, and this addition process provides the network structure of the hydrogels with more sites for the binding of pollutants. Moreover, its use has the advantages of high absorption, efficiency, low cost, easy recovery, and low cost, compared to traditional materials and other methods [107,108].

**Table 2 polymers-14-01498-t002:** Advantages and disadvantages of using adsorption for hydrogels based on chitosan or alginate.

Advantage	Reference	Disadvantage	Reference
Lower toxicity	[103]	Complicated and difficult to handle	[103,104]
Biocompatible	[103]	Non-adherent	[103]
Biodegradable	[105]	Low mechanical and physical strength	[103]
High absorption capacity	[102]	High rate of degradation	[104]
High durability	[107]	Limited versatility	[105]
Stability	[106]		
Environmentally friendly	[108]		
Thermal conductivity	[107]		
Economic	[104]		

After considering the disadvantages and advantages, it can be compared with many similar materials, and we find that we obtain a satisfactory result and a material that absorbs dyes and different charged materials found in liquid media or waste water since the molecular structure of biopolymers helps in the absorption process [109]. It can be used not only for absorption but also for changing the structure of materials, which helps for use in other mechanical removal processes such as with polyelectrolytes in ultrafiltration for dye-removal process from water [110]. In addition, it can be added to several materials as a function of membranes with ion-exchange polymers [111]. This material can participate in several interactions such as the exchange of electrons and π–π, hydrogen bonds, and hydrophobic interactions, and adsorption is the most efficient method (see Table 3) [112] as the hydrogels that have comonomers show a higher density of functional ligands for units to obtain high dye-adsorption capacities that might be changed by different pH levels [113].

## 4. Conclusions

Hydrogels were successfully manufactured using polymers and polysaccharides with high efficiency in removing organic pollutants such as anionic and cationic dyes. The hydrogels were characterized by FTIR spectroscopy, TGA, and their water swelling properties to determine their chemical structure, thermal stability, and water uptake and stability. Using these hydrogels as biobased sorbents to remove dyes from aqueous solution, it was shown that during MB absorption, the best pH was 8.5, with a biomass dosage of up to 0.05 g for an initial dye concentration of 10 mg/L and an absorption time of 10 min using hydrogels with 24% alginate biopolymer content. Using different concentrations up to 1500 mg/L, the absorption efficiency was high, up to 98%. For MO absorption, the best pH was 3.0 using a biomass dosage of up to 0.05 g for an initial dye concentration of 10 mg/L and an absorption time of 10 min. Using 24% hydrogels, chitosan biopolymer, and different concentrations up to 1000 mg/L, the absorption efficiency was as high as 43%. Furthermore, the dye adsorption capacity increased with increasing initial dye concentrations from 10 mg/L to 1500 mg/L for MB and 10 mg/L to 1000 mg/L for MO, and the adsorption capacity decreased for the adsorption of MO with increasing amounts of salts (LiCl, Li_2_CO_3_, CaCO_3_, NaCl, and KCl) but was stable for MB under the same ideal experimental conditions. The study showed that salts of NaCl, KCl, CaCO_3_, Li_2_CO_3_, and LiCl had an effect on the adsorption of MB and MO dyes, and the decrease was due to the competitive effect between MB or MO dyes with salts on the adsorption site. It was observed that adsorption capacity can remain stable and not affected by salts for the MB dye, where the decrease was only 2–10%, but salts can compete with the absorption of MO dye. In the case of MO adsorption, the decrease was sharp, reaching 20–60%. 

This work promotes further development in the design of composite hydrogels with high absorption capacity, high stability, environmental friendliness, and their applications in polluted water treatment.

## Figures and Tables

**Figure 1 polymers-14-01498-f001:**
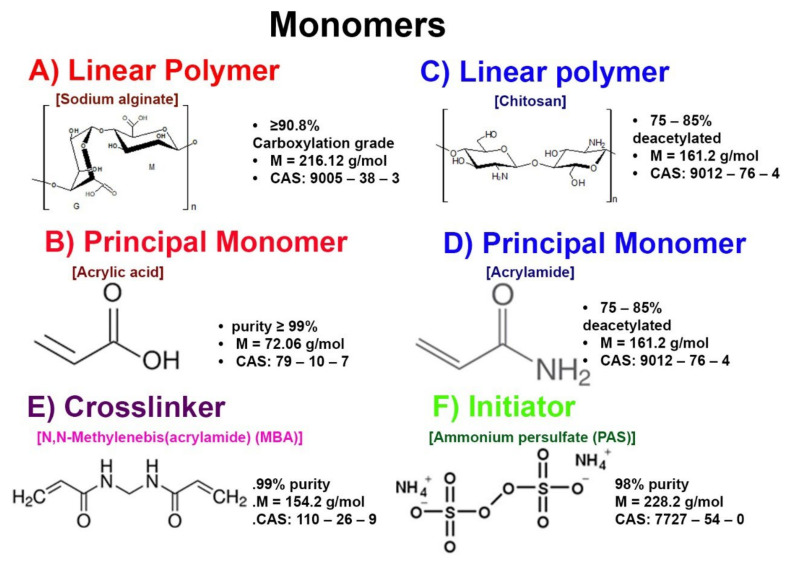
Structures of the monomers and their specifications.

**Figure 2 polymers-14-01498-f002:**
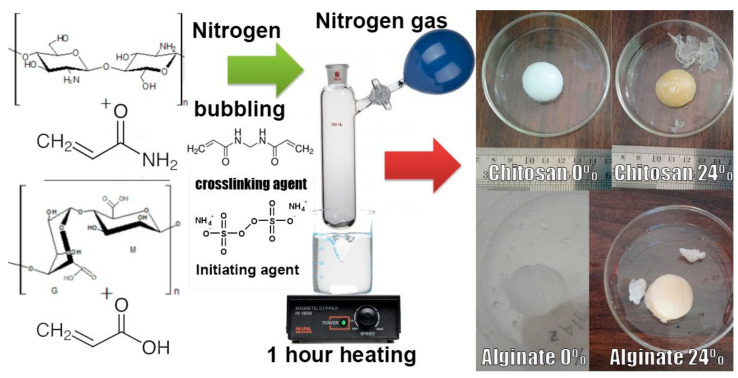
Synthetic scheme of hydrogels using biopolymers and monomers.

**Figure 3 polymers-14-01498-f003:**
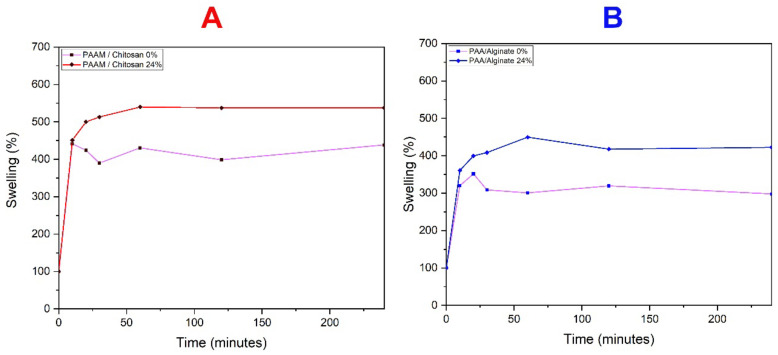
Swelling of (**A**) PAAM/chitosan and (**B**) PAA/alginate xerogels in water.

**Figure 4 polymers-14-01498-f004:**
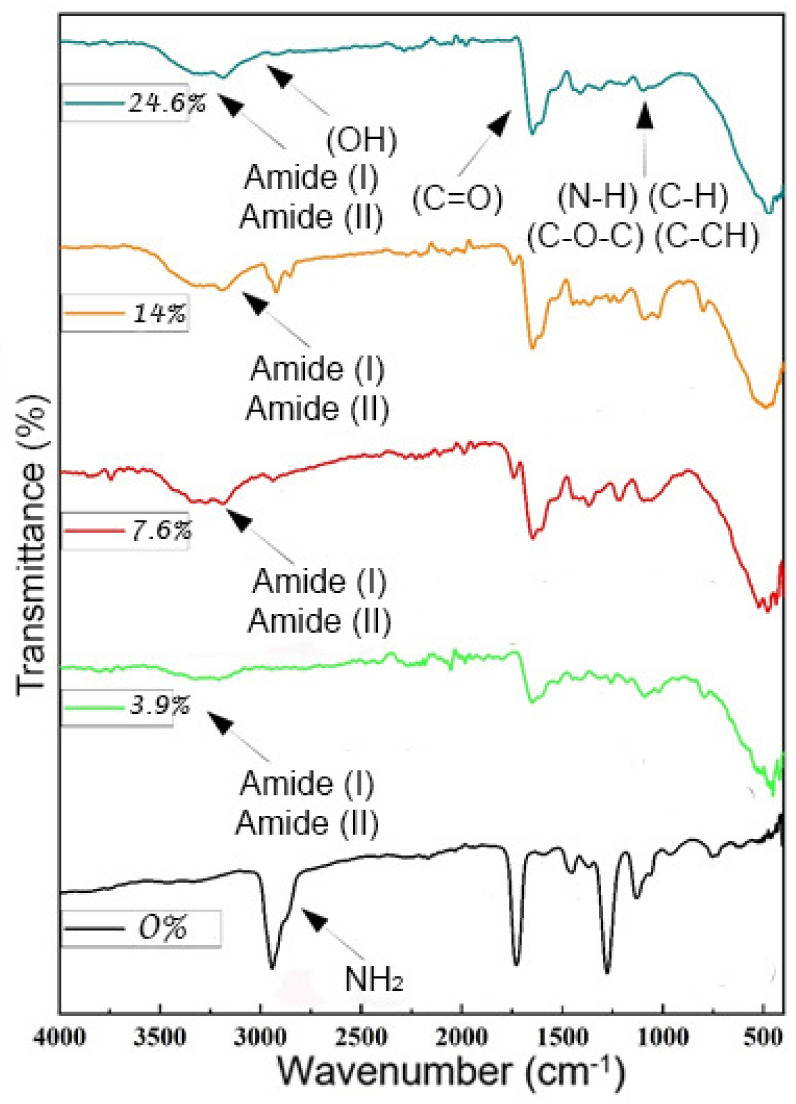
FTIR spectra of PAAM/chitosan xerogels containing different amounts of chitosan.

**Figure 5 polymers-14-01498-f005:**
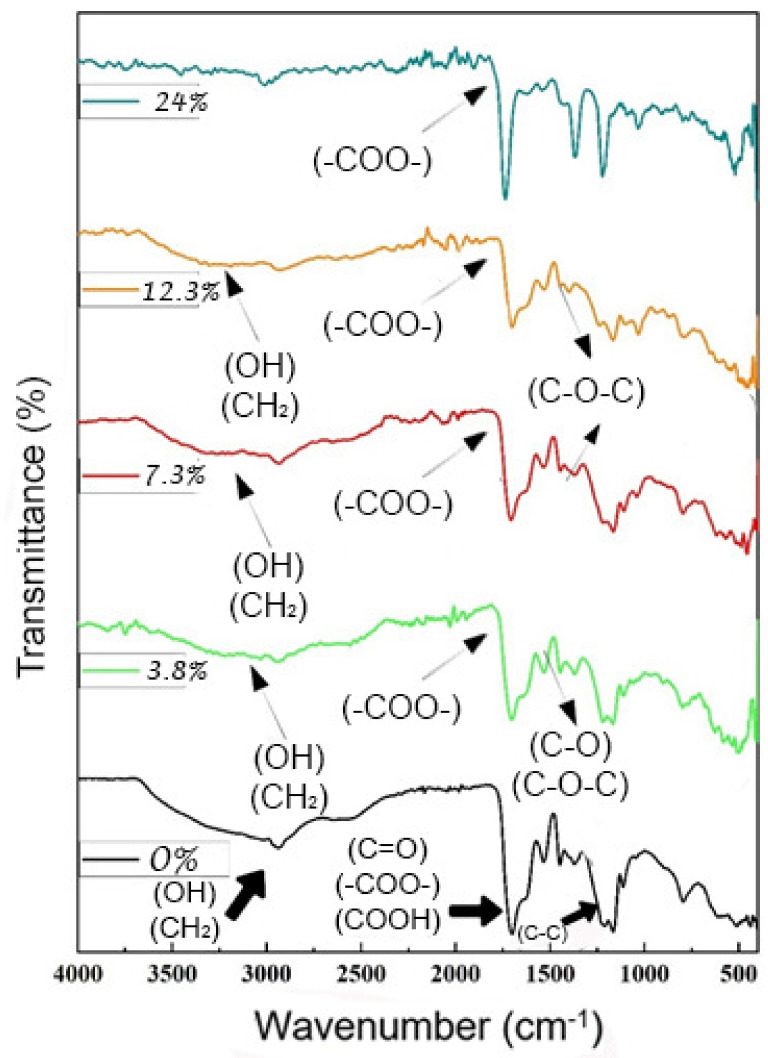
FT-IR spectra of PAA/alginate xerogels containing different amounts of alginate.

**Figure 6 polymers-14-01498-f006:**
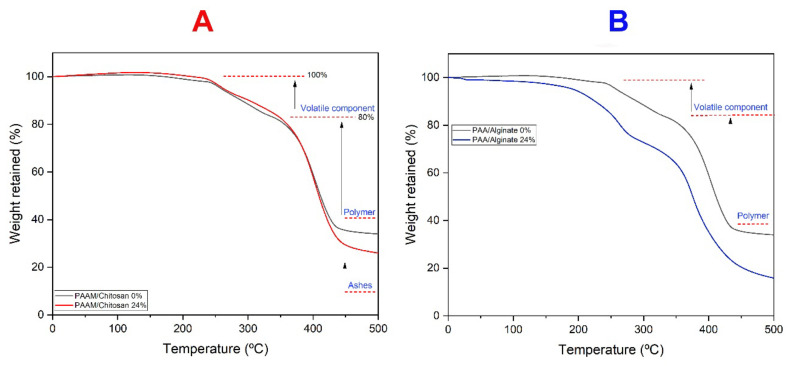
Thermogravimetric analysis (TGA) of (**A**) PAAM/chitosan xerogels and (**B**) PAA/alginate xerogels.

**Figure 7 polymers-14-01498-f007:**
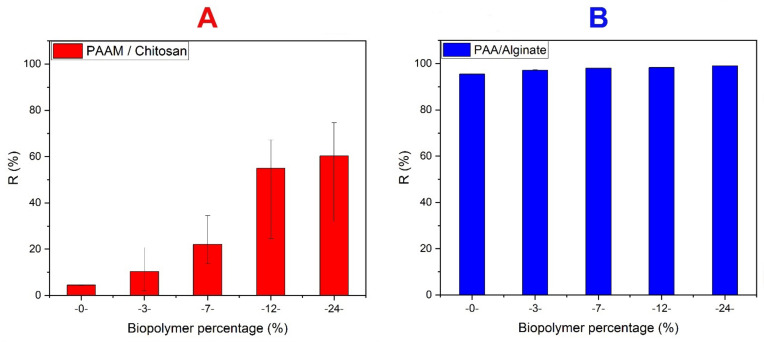
(**A**) Study of the chitosan percentage effect on PAAM/chitosan xerogels adsorbing MO (10 mg/L) at pH 3; (**B**) study of the alginate percentage effect on PAA/alginate xerogels adsorbing MB (10 mg/L) at pH 8.5.

**Figure 8 polymers-14-01498-f008:**
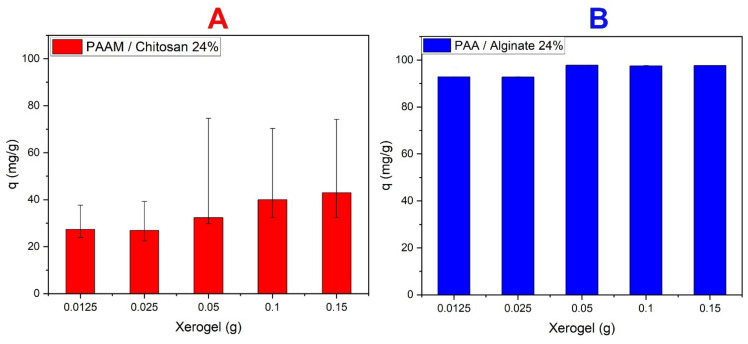
Effect of quantities of xerogels dose (in grams) (**A**) PAAM/chitosan 24% for MO (10 mg/L) removal in pH 3 and (**B**) PAA/alginate 24% for MB (10 mg/L) removal in pH 8.5.

**Figure 9 polymers-14-01498-f009:**
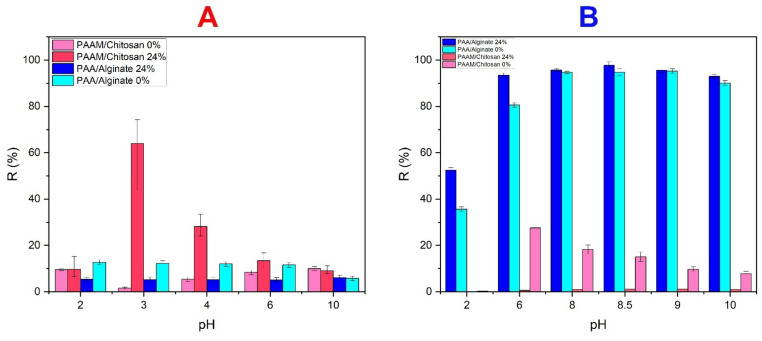
Effect of pH on the adsorption capacity. (**A**) Xerogel PAAM/chitosan (0.05 g) adsorbing MO (10 mg/L). (**B**) Xerogel PAA/Alginate (0.05 g) adsorbing MB (10 mg/L).

**Figure 10 polymers-14-01498-f010:**
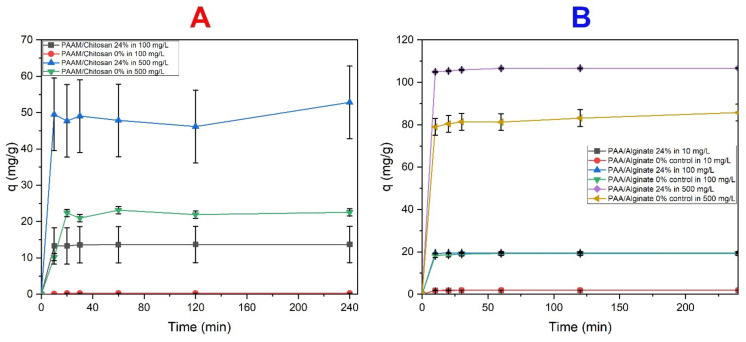
The effect of interaction time on the adsorption capacity using different dye concentrations. (**A**) Adsorption time of PAAM/chitosan (0.05 g) adsorbing MO (100, 500 mg/L) at pH = 3. (**B**) Adsorption time of PAA/Alginate (0.05 g) adsorbing MB (10,100, and 500 mg/L) at pH = 8.5.

**Figure 11 polymers-14-01498-f011:**
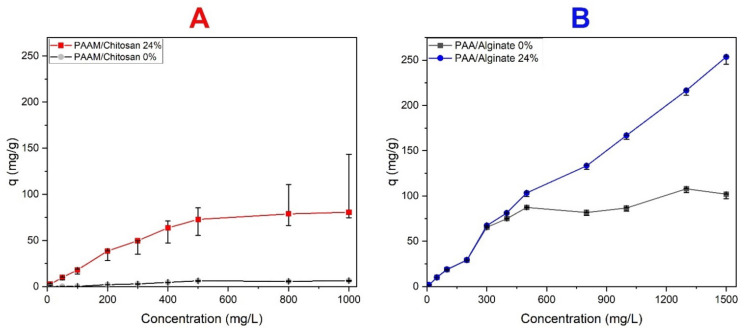
Effect of initial concentration on the adsorption capacity. (**A**) Adsorption capacity of PAAM/chitosan adsorbing MO at pH = 3. (**B**) Adsorption capacity of PAA/Alginate adsorbing MB at pH = 8.5.

**Figure 12 polymers-14-01498-f012:**
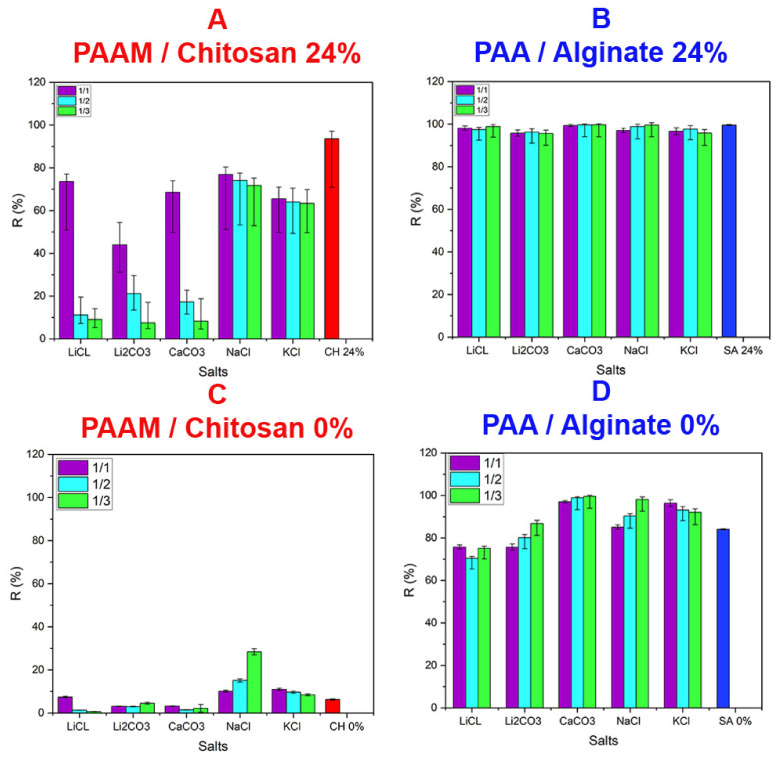
Effect of interfering ions on dye removal for the following salts: LiCl, Li_2_CO_3_, CaCO_3_, NaCl, and KCl, using 1/1, 1/2, and 1/3 molar ratios with dyes. (**A**) Adsorption removal efficiency for PAAM/chitosan 24% xerogel (0.05 g) with MO (100 mg/L) at pH 3.0. (**B**) Adsorption removal efficiency for PAA/Alginate 24% xerogel (0.05 g) with MB (500 mg/L in pH 8.5). (**C**) Adsorption removal efficiency for PAAM/chitosan 0% xerogel (0.05 g) in the presence of salts with MO (100 mg/L) at pH 3. (**D**) Adsorption removal efficiency for PAA/Alginate 0% xerogel (0.05 g) in the presence of salts with MB (500 mg/L) at pH 8.5.

**Table 1 polymers-14-01498-t001:** Details of the amounts of reagents used in each synthesis and its yield (%). MBA was 0.3114 g, and PSA was 0.0116 g.

n° Tube	Percentage (%)	PAA (mL) ± 0.005 mL	Alginate (g) ± 0.005 g	Yields (%)
1	0	1.370 (1.489 g)	0.000	94.3
2	3.8	1.370 (1.489 g)	0.071	96.6
3	7.3	1.370 (1.489 g)	0.142	97.1
4	12.3	1.370 (1.489 g)	0.284	98.6
5	24	1.370 (1.489 g)	0.569	94.4
6	40	1.370 (1.489 g)	1.208	91.3
7	55.66	1.370 (1.489 g)	2.274	95.4
**n° Tube**	**Percentage** **(%)**	**PAAM (g)** **± 0.005 g**	**Chitosan (g) ± 0.005 g**	**Yields (%)**
8	0	1.441	0.000	94.6
9	3.9	1.441	0.072	95.7
10	7.6	1.441	0.144	99.1
11	14	1.441	0.288	99.7
12	24	1.441	0.577	96.0
13	40	1.441	1.176	96.4
14	60	1.441	2.646	94.7

**Table 3 polymers-14-01498-t003:** Chitosan and alginate hydrogels adsorption table.

Material	Dye	Concentration (mg/L)	q (mg/g)	pH	R (%)	Reference
Chitosan microspheres	MO	40	207	3	88	[114]
Chitosan/graphene oxide	MO	800	687	8.45	-	[115]
Chitosan/montmorillonite	MO	40	545	3	-	[116]
Chitosan/ carboxymethyl cellulose—GO	MO	50	405	3	82	[117]
Chitosan/PAAM	MO	10–1000	62.5	3	75	This study
Alginate/GO-montmorillonite	MB	30–200	150.66	5.99	97	[118]
Alginate/polyethyleneimine	MB	100	400	5.5	99	[119]
Alginate/PAA—ZNO	MB	40	1529.6	6	99	[120]
Alginate/PAA	MB	40	1129	6	-	[120]
Alginate/polyaspartate	MB	10	4.85	-	-	[121]
Alginate/PAA	MB	10–1500	120	8.5	98	This study

## Data Availability

All the data are available within the manuscript.

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
