# Peer review of "Chitosan- and Alginate-Based Hydrogels for the Adsorption of Anionic and Cationic Dyes from Water"

_polymers, 2022, doi:10.3390/polym14081498_

Round 1
Reviewer 1 Report
I thoroughly reviewed the manuscript “Chitosan- and alginate-based hydrogels for the adsorption of anionic and cationic dyes from water” by ALSamman and Sanchez, who experimentally studied the the dye removal performances of hydrogels in water made by interpenetrated polymer networks with chitosan and alginate, respectively.
The materials are fairly well characterized and their remediation performances properly described.
However, as described below, I believe that the contents should be more carefully contextualized in the frame of the literature on the topic, and have a few comments that the Authors should consider.
I will be glad to reconsider a revised version of the paper for publication in Polymers, after the Authors will have addressed my concerns and modified the manuscript accordingly.
- The rationale for the unit reported in eq.1, 2 and 4 is not clear to me, since all the defined quantities are in fact adimensional. In particular, while reporting (g/g) in eq.1 and 2 seems to be simply useless, I don’t understand why qt should be expressed in (mg/g) rather than in the more natural adimensional units.
-“where C0 is the solution at the initial time”. Perhaps, C0 is the concentration of “the solution at the initial time”
-“Chitosan and its derivatives have been widely used as sorbents for various types pollutants in water.” This statement should be properly referenced, by mentioning some examples of application of chitosan based materials for water remediation. In addition to hydrogels, for example, also chitosan hydrogels have been also used for puroposes similar to those of this work: e.g. see Int. J. Mol. Sci. 2021, 22, 5535.
-Among the “technological properties” of alginate gels, the Authors should also mention the anomalous aging and stress relaxation characterizing this gels: e.g. see Macromolecules 2020, 53, 649−657 and Polymers 2021, 13, 3618 (a recent Editor’s choice in Polymers).
Author Response
- The rationale for the unit reported in eq.1, 2 and 4 is not clear to me, since all the defined quantities are in fact a dimensional. In particular, while reporting (g/g) in eq.1 and 2 seems to be simply useless, I don’t understand why qt should be expressed in (mg/g) rather than in the more natural a dimensional unit.
Response: Thank you for your observation, the equations in the revised manuscript hve been modified according to the equations in the above paper as an example https://doi.org/10.3390/su13020984
-“where C0 is the solution at the initial time”. Perhaps, C0 is the concentration of “the solution at the initial time”
Response: It is correct , C0 is the solution concentration at the initial time
-“Chitosan and its derivatives have been widely used as sorbents for various types pollutants in water.” This statement should be properly referenced, by mentioning some examples of application of chitosan based materials for water remediation. In addition to hydrogels, for example, also chitosan hydrogels have been also used for purposes similar to those of this work: e.g. see Int. J. Mol. Sci. 2021, 22, 5535.
Response:
Thank you, we added this related research paper
-Among the “technological properties” of alginate gels, the Authors should also mention the anomalous aging and stress relaxation characterizing this gels: e.g. see Macromolecules 2020, 53, 649−657 and Polymers 2021, 13, 3618 (a recent Editor’s choice in Polymers).
Response:
Thank you, we added this information in the revised version of the manuscript.
Reviewer 2 Report
“Chitosan- and alginate-based hydrogels for the adsorption of anionic and cationic dyes from water” manuscript was devoted to the development of hydrogel systems based on PAAM/chitosan or PAA/alginate proposed to reduce the concentrations of dyes in water. The manuscript is interesting, well written, and rich of analysis and experimental details.
However, some major revision need to be performed in order to make it suitable for publication in Polymers:
- In Figure 2 A, please replace carboxilation with carboxylation. In any case, figure 1 appears pleonastic. I suggest to report in the text the information reported in this figure.
- Table 1, in the column PP (mL), since the error in the volume measure was expressed by a number with three decimal digits (i.e., 0.005 mL), please correct the volume values using the same decimal digits number, i.e. 1.370 instead of 1.37 and 1.441 instead of 1.4412. Similarly, in the "Alginate" column, you must round all the numbers to the third decimal place (for example 0.07107 mut be round to 0.071 and so on).
- I suggest using Arial font in all the figures.
- “Removal of dies”: Have you performed all your experiments (reported in Figures 7-12) in replicates, in order to obtain error bars for the histograms or for the points? Even if you state in 123-124 lines "Triplicate experiments were performed for all operating variables studied, and results were reported as the mean values. It was found that the average deviation of the results was low", I suggest reporting the standard deviations as error bars in the all the graphs. This approach should allow obtaining an evaluation of statistically significant differences between the obtained data. Therefore, I suggest carrying out a statistical analysis of the results.
Author Response
In Figure 2 A, please replace carboxilation with carboxylation. In any case, figure 1 appears pleonastic. I suggest to report in the text the information reported in this figure.
Response:
Thank you for your comment, all graphics were modified in the revised version
Table 1, in the column PP (mL), since the error in the volume measure was expressed by a number with three decimal digits (i.e., 0.005 mL), please correct the volume values using the same decimal digits number, i.e. 1.370 instead of 1.37 and 1.441 instead of 1.4412. Similarly, in the "Alginate" column, you must round all the numbers to the third decimal place (for example 0.07107 must be round to 0.071 and so on).
Response:
All numbers have been modified, and the post-comma has also been modified
I suggest using Arial font in all the figures.
Response:
Arial font size 10 was used in the revised figures
“Removal of dyes”: Have you performed all your experiments (reported in Figures 7-12) in replicates, in order to obtain error bars for the histograms or for the points? Even if you state in 123-124 lines "Triplicate experiments were performed for all operating variables studied, and results were reported as the mean values. It was found that the average deviation of the results was low", I suggest reporting the standard deviations as error bars in the all the graphs. This approach should allow obtaining an evaluation of statistically significant differences between the obtained data. Therefore, I suggest carrying out a statistical analysis of the results.
Response:
The average value was used in all studies. A lower value and a higher value were added to graphs.
Reviewer 3 Report
Novel hydrogel systems based on polyacrylamide/chitosan (PAAM/chitosan) or polyacrylic acid/alginate (PAA/alginate) were prepared, characterized and applied to reduce the concentrations of dyes in water.The investigations are interesting and could be published after revision.
- Why the authors use different polymers –polyacrylamide for chitosan (PAAM/chitosan) and polyacrylic acid for alginate (PAA/alginate). Why PAMM could not be teste with both chitosan and then with alginate ? both chitosan and then with alginate could be also tested in polyacrylic acid?
-In the adsorption experiments, methylene blue (MB) was used as a cationic dye, and methyl orange (MO) was used as an anionic dye. Could other dyes be tested in the adsorption experiments ?
-How it could be explained that swelling has a maximum at about 20 min in Figure 3 B?
-At 121 °C, the slight weight loss of 11.2% was attributed to the decomposition of chitosan ? Could authors demonstrate mechanism of the thermal degradation ?
-The authors should describe clearly advantages and disadvantages of the new adsorbents as compared with those of other materials used in this field of applications.
-Could properties of the new adsorbents be compared with other commercially available products used in this field ?
-The advantages and disadvantages should be well described in conclusions.
Author Response
- Why the authors use different polymers –polyacrylamide for chitosan (PAAM/chitosan) and polyacrylic acid for alginate (PAA/alginate). Why PAMM could not be teste with both chitosan and then with alginate ? both chitosan and then with alginate could be also tested in polyacrylic acid?
Response:
Because there is a similarity to the functional groups of NH2 for the chitosan and polyacrylamide, and can be charged with a positive charge and thus lead to an increase in the ability to attract negatively charged elements such as anionic dyes
On the other hand, there are alginates that contain negative functional groups with polyacrylic acid those that also contain a negative charge and that combine with strong electrostatic forces with positively charged dyes.
If we use PAAM with alginate, or PAA with chitosan, probably we will form polycation-polyanion complex with blocked functionalities for dyes removal.
-In the adsorption experiments, methylene blue (MB) was used as a cationic dye, and methyl orange (MO) was used as an anionic dye. Could other dyes be tested in the adsorption experiments ?
Response:
Yes, it can be applied to different charged pollutants, whether it is negatively charged or positively charged such as metals, dyes, pesticides and antibiotics.
-How it could be explained that swelling has a maximum at about 20 min in Figure 3 B?
Response:
It was explained in detail and accurately related to the literature
-At 121 °C, the slight weight loss of 11.2% was attributed to the decomposition of chitosan ? Could authors demonstrate mechanism of the thermal degradation ?
Response:
The TGA book and the analytical book for Thermogravimetric analysis will be reviewed
-The authors should describe clearly advantages and disadvantages of the new adsorbents as compared with those of other materials used in this field of applications.
Response:
A paragraph and table of advantages and disadvantages were added
-Could properties of the new adsorbents be compared with other commercially available products used in this field ?
Response:
Yes, these materials were compared with other types of commercial gels with different absorption mechanisms
-The advantages and disadvantages should be well described in conclusions.
Response:
The advantages and disadvantages were added and discussed properly
Round 2
Reviewer 1 Report
I am satisfied with the revisions, apart from one issue that is probably due to a typo in my previous report.
In particular, I deem Authors should mention in the introduction that also chitosan areogels (I erroneously wrote hydrogels in my previous report) have been also used for purposes similar to those of this work: see Int. J. Mol. Sci. 2021, 22, 5535.
Author Response
Thank your observation. The mentioned paper was included in the revised version.
Reviewer 2 Report
The authors correctly carried out the suggested improvements to the manuscript.
Author Response
thank you for your constructive comments.
Reviewer 3 Report
If editor and other reviewers agree that the paper is suitable for this journal I would recommend the paper for publication after the revision.
Author Response

(The authors gave the same response as above.)
